# Phenotyping of Corn Plants Using Unmanned Aerial Vehicle (UAV) Images

**Wei Su** [1,2], **Mingzheng Zhang** [1,2], **Dahong Bian** [3], **Zhe Liu** [1,2,*], **Jianxi Huang** [1,2], **Wei Wang** [1,2], **Jiayu Wu** [1,2] **and Hao Guo** [1,2]

1   College of Land Science and Technology, China Agricultural University, Beijing 100083, China
2   Key Laboratory of Remote Sensing for Agri-Hazards, Ministry of Agriculture and Rural Affairs, Beijing 100083, China
3   College of Agronomy, Hebei Agricultural University, Baoding 071001, China
*   Correspondence: liuz@cau.edu.cn; Tel.: +86-10-6273-7855

**Abstract:** Phenotyping provides important support for corn breeding. Unfortunately, the rapid detection of phenotypes has been the major limiting factor in estimating and predicting the outcomes of breeding programs. This study was focused on the potential of phenotyping to support corn breeding using unmanned aerial vehicle (UAV) images, aiming at mining and deepening UAV techniques for comparing phenotypes and screening new corn varieties. Two geometric traits (plant height, canopy leaf area index (LAI)) and one lodging resistance trait (lodging area) were estimated in this study. It was found that stereoscopic and photogrammetric methods were promising ways to calculate a digital surface model (DSM) for estimating corn plant height from UAV images, with $R^2 = 0.7833$ ($p < 0.001$) and a root mean square error (RMSE) = 0.1677. In addition to a height estimation, the height variation was analyzed for depicting and validating the corn canopy uniformity stability for different varieties. For the lodging area estimation, the normalized DSM (nDSM) method was more promising than the gray-level co-occurrence matrix (GLCM) textural features method. The estimation error using the nDSM ranged from 0.8% to 5.3%, and the estimation error using the GLCM ranged from 10.0% to 16.2%. Associations between the height estimation and lodging area estimation were done to find the corn varieties with optimal plant heights and lodging resistance. For the LAI estimation, the physical radiative transfer PROSAIL model offered both an accurate and robust estimation performance both at the middle ($R^2 = 0.7490$, RMSE = 0.3443) and later growing stages ($R^2 = 0.7450$, RMSE = 0.3154). What was more exciting was that the estimated sequential time series LAIs revealed a corn variety with poor resistance to lodging in a study area of Baogaofeng Farm. Overall, UAVs appear to provide a promising method to support phenotyping for crop breeding, and the phenotyping of corn breeding in this study validated this application.

**Keywords:** phenotyping; unmanned aerial vehicle (UAV); remote sensing; corn plant height; lodging; LAI

## 1. Introduction

Phenotyping is the process of rapidly profiling crop phenotypic traits such as plant height, canopy cover, density, biomass, and yield [1–3]. The phenotypic traits of interest include geometric traits (e.g., plant height, leaf area index (LAI)), physiological traits (e.g., contents of chlorophyll and other pigments), indicators of abiotic and biotic stress (e.g., canopy temperature differences, leaf water potential), nutrient contents (e.g., nitrogen and protein contents), and yield [4–7]. The performance of crop breeding in terms of its ability to improve crop yield and productivity must be evaluated under natural conditions to account for these factors. However, field breeding trial measurements are

time-consuming and expensive, since they require hundreds to thousands of plots. In addition, the measured results using traditional manual methods are subjective, with measurement error. Therefore, researchers and breeders have been very interested in developing high-throughput phenotyping techniques to overcome these bottlenecks and predict the outcomes of breeding programs rapidly, accurately, and inexpensively. Fortunately, unmanned aerial vehicles (UAVs) are an effective way to acquire crop information rapidly and nondestructively due to their rapid and relatively low cost [5,8]. Compared to satellite remote sensing technologies [9], UAVs provide better spatial resolution and better temporal resolution. Therefore, they have the potential to improve the identification of desirable traits and reduce the risk of data loss due to cloud/raining/smog cover and limitations resulting from the long revisit periods of satellites [10], since UAVs generally operate below the clouds. In addition, UAVs offer lower operating costs and operational complexity than piloted airborne technologies. Therefore, remote sensing with UAVs will be an increasingly important and indispensable tool for phenotyping to support genomics-assisted plant breeding [11].

UAV images have obvious improvements in spatial, spectral, and temporal resolution compared to satellite images [12]. Thus, phenotyping based on UAV images is being rapidly developed [13]. Yang et al. [5] reviewed the advances in field-based phenotyping using UAVs. They summarized the key traits that support crop breeding, such as geometric traits, phenotype-related spectral indices, crop physiological traits, crop abiotic and biotic stress symptoms, nutrient status, and crop yield. Specifically, they estimated corn plant height using UAV-based LiDAR point clouds, and a crop height model (CHM) from LiDAR was created for the corn plant height estimation in their study. Corn lodging estimation was reviewed using the spectral and textural difference between the lodging area and nonlodging area. In addition, the LAI in soybean breeding was estimated using UAV-based hyperspectral images, and the correlations between the triangle ratio vegetation index (TVI), vegetation index (RVI), normalized difference vegetation index (NDVI), and renormalized difference vegetation index (RDVI) with in situ-measured LAI were used to estimate soybean canopy LAI in their study. Crop height [14], LAI [15], and lodging [13] are common, intuitive crop phenotypic traits, as they are important phenotypic traits for crop breeding. These traits can be obtained rapidly by analyzing the spectral, textural, and structural information contained in UAV remote sensing images.

Crop height is defined as the shortest distance between the upper boundary of the main photosynthetic tissues on crop plants and the ground surface [10], which is a good proxy of biomass. Shi et al. [1] estimated corn and sorghum plant heights using the steric UAV images. Their UAV images were collected using a fixed-wing Anaconda UAV (Ready Made RC, Lewis Center, Ohio) and an X88 rotary-wing UAV. Holman et al. [10] utilized a structure from motion (SfM) photogrammetric method to produce 3D topographic reconstructions of a wheat field and derive wheat heights. Their method was able to produce measures of height comparable in accuracy to those of manually measured heights. Watanabe et al. [11] calculated the 50th, 75th, 90th, and 99th height percentiles for a DSM derived from an orthomosaic of UAV images to estimate sorghum plant height. They also found that height estimation performance using UAV remote sensing was similar to that of traditional measurements in genomic prediction modeling. Chapman et al. [13] identified the proportion of wheat lodging using the estimated crop height calculated from a DEM. This application is crucial to breeders because lodging resistance is an important inherited characteristic.

The LAI is defined as the total one-sided leaf area per unit ground surface area for flat broad leaves or half of the total light-intercepting area per unit ground surface area for non-flat leaves [16]. Because of its importance for describing photosynthetic characteristics, the LAI is one of the most common phenotypic traits for characterizing crop growth, energy interception, and many other physical processes of crop growing [17]. Many studies of LAI retrieval or estimation using satellite images have been done, but the spatial resolution is too coarse to support plant breeding. UAVs provide a convenient and efficient way to collect remote sensing images with high spatial and temporal resolution [18]. There are two kinds of popular LAI retrieval methods based on UVV images, including statistical approaches based on vegetation indexes (VIs) [19] and radiative transfer models such as the

PROSAIL model [12,20,21]. The vegetation indexes method is simple and convenient. However, they need numerous in situ-measured data. In addition, the vegetation indexes will be statured when the crop canopy LAI is high [22].

Alternatively, radiative transfer models may be a more robust approach to characterize crop canopy difference. The PROSAIL model is one of the most popular models for LAI retrieval [23,24]: it is a coupling of the leaf level PROSPECT model and the canopy level SAIL model [25]. The key for LAI retrieval using the PROSAIL model is finding the optimal math between simulated crop canopy reflectance through the PROSAIL model and remote sensing image reflectance using a look-up table (LUT) by minimizing the cost function [26]. The limitation of LAI retrieval using the PROSAIL model lies in the ill-posed problem resulting from model uncertainties [27]. Duan et al. [12], Shi et al. [1,6], Roosjen et al. [20], and Xu et al. [21] have studied LAI retrieval and estimation using UAV remote sensing images. Duan et al. [12] used the PROSAIL model to evaluate the suitability of LAI estimation using hyperspectral images. They found that the retrieved LAI was accurate, with a root mean square error (RMSE) of approximately 0.62 $m^2$ $m^{-2}$ and a relative RMSE of approximately 15.5%. Shi et al. [1,6] found a strong relationship ($R^2 = 0.93$) between the UAV NDVI and the in sit- measured LAI of winter wheat. They both concluded that UAV remote sensing images could be used to estimate the LAI of winter wheat. Roosjen et al. [20] retrieved a potato crop canopy LAI using multiangle spectral UAV images, aiming at alleviating the ill-posed problem through multiangle imaging approaches. Xu et al. [21] retrieved a rice canopy LAI based on UAV images by coupling the PROSAIL model and Bayesian network models. In our previous work, we tried to alleviate this problem through the use of a priori information measured in fieldwork and improved PROSAIL inputs, such as the leaf angle distribution function [28]. In this study, the potential of LAI retrieval in the entire growing season of corn using VIs and the PROSAIL model was analyzed, aiming at finding the optimal LAI retrieval method for the entire corn growing season.

Corn (*Zea mays*) is planted widely around the world and is a key component of food security both for humans and for animals [29]. The phenotyping of corn would provide important support for corn breeding to increase the yield of corn. Therefore, this study focused on finding methods for phenotyping of breeding corn plants using UAV images. Our objectives were as follows:

1. Determine if the SfM photogrammetry method can be used to estimate corn plant height based on stereoscopic UAV images and if it can be used to detect height differences and height variations for different corn cultivars;

2. Compare the corn lodging area estimation accuracy using the differences in textural features and the differences in plant height between lodging and nonlodging areas. Corn plant heights were calculated from normalized DSM (nDSM) data calculated from stereoscopic UAV images using SfM photogrammetry;

3. Explore the potential for LAI retrieval using our improved PROSAIL radiative transfer model based on UAV images.

## 2. Materials and Methods

### 2.1. Materials

#### 2.1.1. Study area

There were two corn breeding trials analyzed in this study: the Shunyi corn breeding trial located in Beijing, China, and the Baogaofeng Farm located in Mazhuang Town, Xinji City, Hebei Province, China. Figure 1 shows the locations of these two study areas. The Shunyi corn breeding trial is in northern China (40°11′44.82″N, 116°33′59.49″E) at an elevation of 48 m and covers 20 ha. The region has a warm-temperate semihumid continental monsoon climate with an annual average rainfall of 625 mm, with a dry winter and annual sunshine duration of 2750 hours. The temperature in corn growing season ranges from 7.3 °C to 39.6 °C. The corn growing season is commonly from early June to early September. The UAV flights occurred on 21 July 2016 and 14 September 2016. The UAV images

acquired on 21 July and 14 September were used to estimate the corn plants' heights and lodging areas, respectively, which are the jointing stage and milking stage of corn (individually) in the study area. There were 4 kinds of corn cultivars in field No. 12 of the Shunyi corn breeding trial: they were YY-1, YY-2, YY-3, and QH50, and there were 215 plots in this field. The planting density in field No. 12 was 7500 plants/ha. For field No. 14, there were 324 plots in total, and the corn cultivars included JK-A, JK-B, JK-C, LP-A, LP-B, LP-C, HN-101, HN-803, CY480, LY410, and JNK728. There were four kinds of planting densities in field No. 14, i.e., 60,000 plants/ha, 67,500 plants/ha, 75,000 plants/ha, and 82,500 plants/ha.

The Baogaofeng Farm (37°47′55.26″N, 115°18′02.51″E) is in Mazhuang Town, Xinji City, Hebei Province, which is to the southwest of Beijing at an elevation of 14 m and covers 20 ha. The climate is a semihumid continental climate in a warm monsoon temperate zone with an annual average rainfall of 488.2 mm with a hot rainy season. Precipitation from June to August accounts for 67.9% of the total annual precipitation in Baogaofeng Farm. The corn is sowed at the end of June, the jointing stage is around the middle of July, the flare opening stage is around the end of July, the heading stage is around the middle of August, and the milking stage is around early September. According to the phenological stage of corn in Baogaofeng Farm, UAV image collection and an in situ measurement campaign were carried out on 15 July (the jointing stage), 26 July (the flare opening stage), 11 August (the heading stage), and 4 September (the milking stage) 2018. There were seven kinds of corn cultivars and seven planting densities in Baogaofeng Farm. The corn cultivars involved Zhengdan 958, Xianyu 335, JNK 728, Denghai 605, Xianyu 047, MC 812, and Jinhai 5. The plant densities included 60,000 plants/ha, 67,500 plants/ha, 75,000 plants/ha, 82,500 plants/ha, 90,000 plants/ha, 105,000 plants/ha, and 120,000 plants/ha in total.

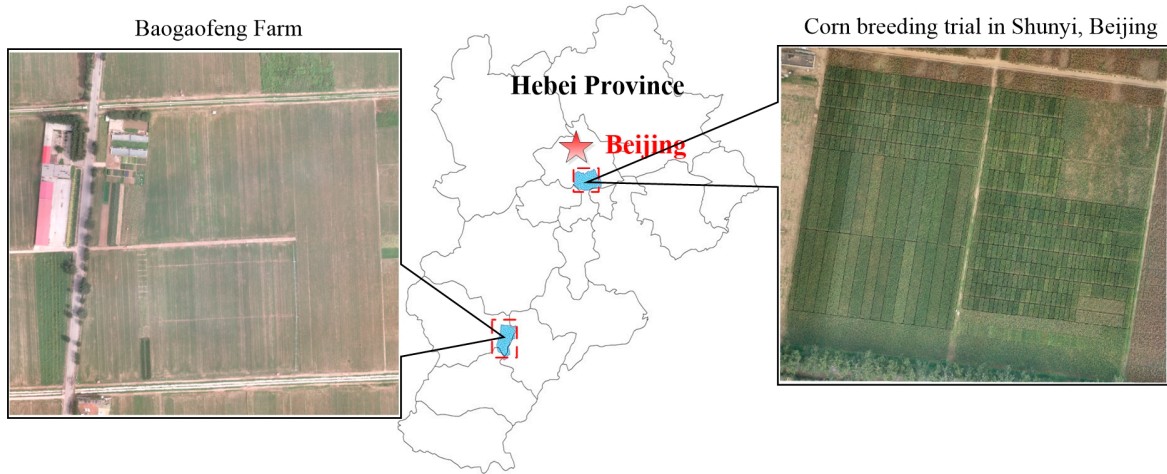

**Figure 1.** Locations of the two study areas. The Baogaofeng Farm study was performed in Mazhuang Town, Xinji City, Hebei Province (left), and the Shunyi corn breeding trial was performed in Shunyi District, Beijing, China (right).

2.1.2. UAV Image Collection

A DJI S1000+ UAV (DJI-Innovations Co., Ltd., Shenzhen, China; Figure 2a) was used to obtain UAV images. The UAV had an onboard GPS receiver with a 2D positioning accuracy of 10 cm. The sensors mounted on the UAV were a Sony DSC QX100 digital camera (Sony Inc., Tokyo, Japan) and a Parrot Sequoia multispectral digital camera (Parrot SA., Paris, France). The UAV had a maximum flight time of 12 min before it was necessary to replace the battery. The Sony DSC QX100 is an RGB camera with a 20.2-megapixel resolution (sensor size of 13.2 mm × 8.8 mm) and an equivalent focal length of 28 to 100 mm. The Parrot Sequoia camera captures green, red, red-edge, and near-infrared band reflectance with a 120-megapixel resolution. The UAV flights were conducted at a height of 50 m above the ground at a travel speed of 6 m/s. Table 1 summarizes the characteristics of the cameras.

In order to obtain reliable corn plant heights, an overlap of over 5 images was set for every pixel for UAV image acquisition. Figure 2b shows the flight paths used to acquire images in this study. To geometrically correct the images, the positions of obviously stationary objects were selected as ground control points (GCPs), with their positions determined to an accuracy of 8 mm using the GPS receiver. There were 21 GPS control points set in Baogaofeng Farm, and there were 36 GPS control points set in the Shunyi corn breeding trials. These GPS points were distributed throughout the study area as evenly as possible. Obvious objects such as the corner of a room for storing motor-pumped wells in Baogaofeng Farm, the corner of a room for storing agricultural machinery in Shunyi, and ridges of corn fields were commonly used as GPS points.

**Table 1.** Characteristics of the cameras used to obtain the unmanned aerial vehicle (UAV) images.

| Parameters | Parrot Sequoia Multispectral Camera | Sony DSC QX100 Camera |
|---|---|---|
| Type | Multispectral | Visible RGB |
| Weight (g) | 72 (camera) + 36 (light intensity sensor) | 300 |
| Spectral bands | Green, red, red-edge, near-infrared, visible | Blue, green, red |
| Effective pixels | 1.2 Mpx (multispectral bands), 16 Mpx (visible bands) | 20.2 Mpx |

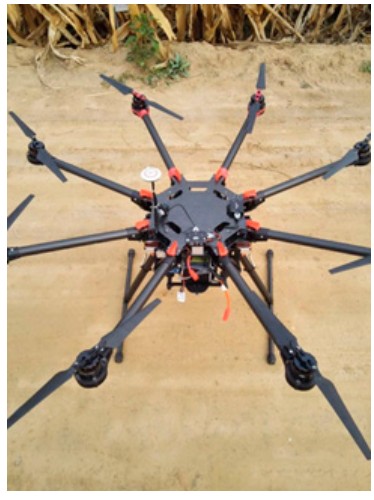 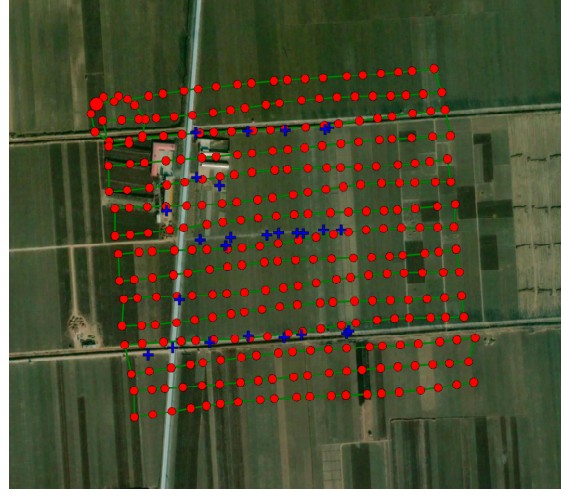

(**a**) UAV platform          (**b**) UAV flight path

**Figure 2.** The DJI S1000+ UAV (**a**) used in this study and (**b**) an example of its flight path. Note: red points are the location of pictures taken by the UAV, and the blue plus signs are the locations of ground GPS.

2.1.3. UAV Image Preprocessing

UAV image preprocessing was done after each flight, including an image mosaic, geometric correction, and the creation of orthomosaics using Pix4Dmapper software (https://pix4d.com/). The geometric correction was done to correct for any flight instability and errors in the airborne GPS recording. Other preprocessing included radiometric calibration to convert the pixel values from digital numbers to reflectance values using the measured reflectance for all spectral bands. Radiometric calibration used vicarious calibration based on the absolute reflectance method [30] using the radiation calibration board in Baogaofeng Farm. The radiation calibration board was placed on a flat ground surface in the study area, and it was ensured that there were no shadows covering the radiation calibration board (shown as Figure 3a). Figure 3b–e represents the captured green, red, red-edge, and NIR band images for radiometric calibration in Baogaofeng Farm, Xinji City, Hebei Province, China. These pictures taken of the radiation calibration board were added to Pix4Dmapper software for radiometric calibration during the image mosaic process.

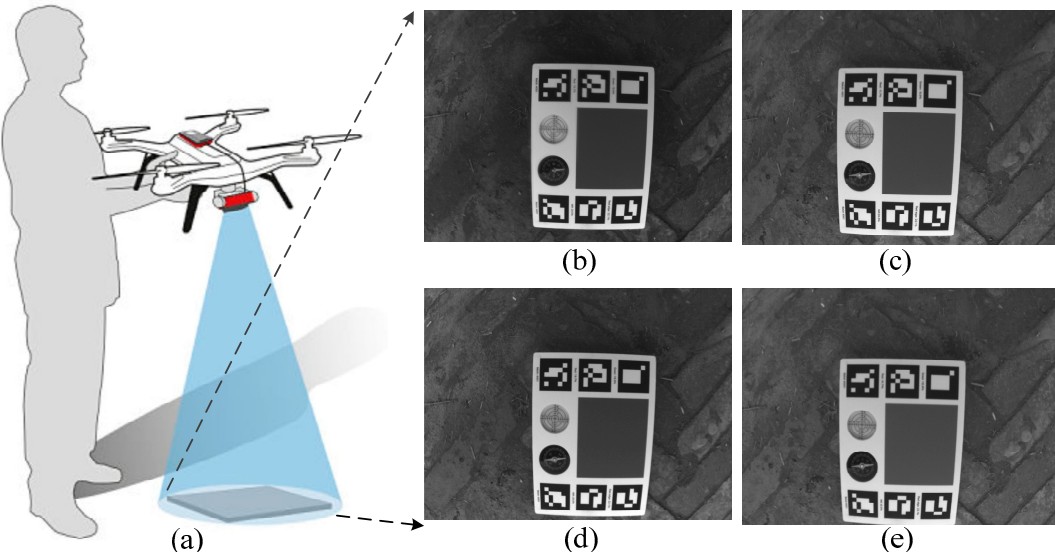

**Figure 3.** Sketch of radiometric calibration (**a**) and captured green (**b**), red (**c**), red-edge (**d**), and NIR (**e**) band images for radiometric calibration in the study area.

Furthermore, a points cloud was produced using Pix4D software for corn plant height modeling. For generating the expected height accuracy, the parameters were set in the Pix4D software as follows: The pixel matching was 1, which could produce points and depict more corn plant structural details. Next, a multiscale was ticked, which ensured that there were more densified 3D points and more details of corn plants, too. In addition, the point density was set to high with a slow speed, and the minimum number of matches per 3D point was 4.

### 2.1.4. Field Campaign

Synchronous with the UAV flights, in situ measurements of the corn plant heights, lodging area, canopy LAI, leaf chlorophyll content, and spectral reflectance of the corn canopy were done at each site. The locations of the sample sites were obtained using a Huace i80 real-time kinematic (RTK) GPS receiver (Huace Ltd., Shanghai, China). The corn plant heights were measured using a meter stick from the ground to the tip of the corn plants. The measured plant heights were used to validate the estimated height of a pixel at its location. The lodging areas were located using a Huace i80 real-time kinematic (RTK) GPS receiver, too. The lodged areas were labeled to validate the lodging area estimation. The corn canopy LAI was measured using an LAI-2200 Plant Canopy Analyzer (LI-COR, Lincoln, NE, USA), and the cover cap was set to a 45° field angle to eliminate the effect of nonplant objects within the range of the sensor's field of vision. There were six measurement values that were acquired in each plot following a transect with a zigzag pattern. The chlorophyll content of the corn leaves was measured using a SPAD-502 leaf chlorophyll meter (Konica Minolta Corp., Solna, Sweden). There were five corn plants that were selected randomly in each plot, and the chlorophyll content of six leaves per plant was measured. For each leaf, we measured the chlorophyll content near the leaf tip, leaf bottom, and middle of the leaf. We used the average of these measurements to represent the chlorophyll content for each leaf and the average for all leaves to represent the content in each plot. The reflectance of the corn canopy, the white diffuse reflector, the black diffuse reflector, and the white cloth were measured using an SVC HR-1024 spectroradiometer (Spectra Vista Corporation, Poughkeepsie, NY, USA), and the data were used for radiometric calibration of the UAV remote sensing images.

### 2.2. Methods

There were three phenotypic parameters that were estimated for breeding corn plants using UAV images in this study: plant heights, lodging area, and canopy LAI. During the phenotyping process,

the plant heights were estimated using the difference in the DSM coming from the stereo UAV images through the SfM method and the DTM by means of ordinary Kriging interpolation of bare earth points, i.e., nDSM. For the lodging area estimation, there were two kinds of methods used: textural features and the calculated nDSM. For the corn canopy LAI, the physical radiative transfer PROSAIL model was used.

### 2.2.1. Corn Plant Height Estimation

Corn plant height is a morphological trait and is expressed as the shortest distance between the upper boundary of the main photosynthetic tissues of a plant and the ground level [10]. To estimate plant heights, a DSM characterizing the height of ground and corn plants was created in Pix4D software using generated 3D points. The 3D points were created from stereo UAV images using the SfM method. There were overlapping regions between adjacent UAV images. To account for these overlaps, all of the UAV images and their position and orientation system (POS) data (acquired at the same time) were used to calculate the exterior elements of adjacent UAV images using aerial triangulation. Next, a digital terrain model (DTM) characterizing the height from the ground only was produced using points located on the bare ground. The bare ground points were selected based on the DSM image and true-color RGB UAV images using the hierarchical moving curve-fitting method [31]. Then, intermediate points were generated in the DTM by means of ordinary Kriging interpolation. Last, the nDSM was generated as the difference between the DSM and DTM elevations, characterizing the corn plant height pixel by pixel. These three steps, including the selection of bare ground points, interpolation, and a calculation of the difference between the DSM and DTM, were completed using our custom code.

### 2.2.2. Lodging Area Estimation

Lodging is a major cause of corn yield loss and occurs as a result of extreme weather (e.g., strong winds, heavy rain) or improper management (such as maintaining excessively high levels of soil moisture) [3]. Thus, lodging resistance is an important genetically determined characteristic that is an important criterion in corn breeding. The lodging area in the Shunyi corn breeding trial was estimated using 2 kinds of methods: mining the textural differences or the height differences between corn plants with and without lodging.

The textural differences between lodging and nonlodging areas were used to estimate lodging areas first. The gray-level co-occurrence matrix (GLCM) is widely used to extract textural features from high-resolution remote sensing images [32], and it was developed by Haralick et al. [33]. The GLCM textural features of UAV images were calculated in the spatial domain to identify lodging areas in this study. The gray-level co-occurrence matrix stores the probability of co-occurrence between two gray levels, $i$ and $j$, and the given relative orientation ($y$) and distance ($d$) [32]. Therefore, the essentials of the GLCM method lie in the determination of the window size, the offset value, the channel, and which feature to use. There were 14 original textural features developed by Haralick et al. [33], including the mean, variance, homogeneity, contrast, dissimilarity, entropy, angular second moment (ASM), correlation, etc. These features are shift-invariant and are effective texture discriminators. Because this study did not focus on how to set reasonable values for these parameters, the details of decisions on window sizes, offset values, the channel, and which features to use were referenced from principles in our previous work [34]. In this study, ASM textural features were extracted based on the blue, green, and red bands of the Sony DSC QX100 digital camera pictures, and a $5 \times 5$ window size and an offset value of 1 were used simultaneously. The ASM computation was done in ENVI software (version 5.3). Based on the textural differences between the lodging and nonlodging areas, the lodging areas were classified using an SVM-supervised classifier in ENVI software (version 5.3) by selecting training samples.

The second method used height differences between the nDSMs in lodged and unlodged areas. And the height differences were determined by the thresholds, which were captured by statistics of lodging and nonlodging areas in the study area.

### 2.2.3. LAI Retrieval Using the PROSAIL Model

Compared to statistical and empirical vegetation index approaches, physical radiative transfer models have shown great flexibility in retrieving crop LAIs because they can be parameterized for a wide range of land cover situations and sensor configurations. The PROSAIL model is a popular model for retrieving LAIs [24]. It was developed by coupling the SAIL bidirectional canopy-reflectance model [35] with the PROSPECT leaf optical properties model [36,37]. PROSPECT pioneered the simulation of directional-hemispherical reflectance and transmittance [38] using the leaf structure parameter $N$ and leaf biochemical content variables (e.g., chlorophyll $a + b$ content ($C_{ab}$), carotenoid content ($C_{ar}$), and equivalent water thickness ($C_w$)) at the leaf level. The simulated reflectance and transmittance are used as inputs for the SAIL model to retrieve the LAI using canopy inputs such as measured LAI, the leaf inclination distribution function (*LIDF*), a hot spot parameter (*hspot*), the solar zenith angle ($\theta_s$), the viewing zenith angle ($\theta_v$), the relative azimuth angle ($\varphi_{sv}$), soil reflectance (assumed to be Lambertian or not; $\rho_{soil}$), and the ratio of diffuse to total incident radiation (*skyl*) [39,40]. The forward simulation process uses the following model:

$$\rho_c = \text{PROSAIL } (N, C_{ab}, C_{ar}, C_w, C_m, LAI, LIDF, hspot, \theta_v, \theta_s, \varphi_{sv}, \rho_{soil}, skyl), \tag{1}$$

where $\rho_c$ is the canopy reflectance. There are four kinds of inputs for the PROSAIL model: leaf optical properties ($N$, $C_{ab}$, $C_m$, $C_w$), canopy structural properties (*LAI*, *LIDF*, hspot), soil and sky properties ($\rho_{soil}$ and *skyl*), and sun sensor properties ($\theta_s$, $\theta_v$, $\varphi_{sv}$). Among these parameters, the leaf chlorophyll ($C_{ab}$) was measured with a SPAD-502 chlorophyll meter in this study. Equivalent water thickness ($C_w$) is tied to the difference between fresh leaf weight and dry leaf weight ($C_w = (C_{fresh\ leaf} - C_{dry\ leaf})/LAI$). The LAI came from the measured LAI value in fieldwork using an LAI-2200 Plant Canopy Analyzer [41,42]. The values of $\theta_s$, $\theta_v$, and $\varphi_{sv}$ were obtained from the acquisition parameters for UAV image collection. These parameters were used to generate the LUT in order to reduce the ill-posed problem of the vegetation canopy parameter retrieval.

For the PROSAIL input observations, *LAI*, *LIDF*, $C_{ab}$, $C_m$, $C_w$, $\theta_s$, and $\theta_v$ were sensitive inputs within blue, green, red, and NIR bands, which meant that there would be obviously different simulated reflectance when these inputs changed. Therefore, the ranging values and distributions (i.e., uniform or Gaussian) for these inputs were set (Table 2). Especially, the *LIDF* was a sensitive input for corn canopy LAI retrieval, which was revealed in our previous work [39]. Therefore, we improved the *LIDF* by refining the corn leaf inclination distribution function using terrestrial LiDAR data. The inferred maximum probability leaf angles were used in the Campbell ellipsoid leaf angle distribution function of PROSAIL. This was our improvement of PROSAIL model used for corn canopy LAI retrieval in Baogaofeng Farm using four UAV images acquired on 15 July, 26 July, 11 August, and 4 September. The canopy reflectance forward simulation was done by coupling the leaf-level PROSPECT model and the canopy-level SAIL model. By inputting the leaf-leveled $N$, $C_{ab}$, $C_{ar}$, $C_{brown}$, $C_m$, and $C_w$ values listed in Table 2, the directional-hemispherical reflectance and transmittance of a leaf [38] over the spectrum from 400 nm to 2500 nm [36] of the leaf were simulated. Coupling the leaf reflectance and transmittance, the canopy inputs, including LAI, LIDF, and hspot, were used to simulate canopy reflectance. The code of forward PROSAIL Python Bindings was downloaded from https://pypi.org/project/prosail/, and the codes on backward LAI retrieval using the PROSAIL model and the improvement of the corn leaf inclination distribution function were our custom codes. In the next step, the best fit between the simulated canopy reflectance and the observed canopy remote sensing reflectance was determined using a look-up table (LUT). By finding the best fit using a LUT, the corn canopy LAI could be retrieved. A cost function based on the root mean square error (RMSE) was used to quantify the difference

between the simulated reflectance in the LUT and the observed reflectance for multiple bands of the remote sensing images, which was computed as

$$\text{RMSE} = \sqrt{\frac{1}{n}\sum_{\lambda=1}^{n}(R_{\text{sim}}(\lambda) - R_L(\lambda))^2} \tag{2}$$

where $R_{\text{sim}}(\lambda)$ is the simulated reflectance of band $\lambda$ in the LUT, $R_L(\lambda)$ is the UAV image reflectance in band $\lambda$, and $n$ is the number of wavelength bands. The retrieved LAI is found when the RMSE approaches 0.

**Table 2.** The ranges and distributions of the PROSAIL model inputs for LAI retrieval.

| | | Model Variables | Range or Value | Distribution |
|---|---|---|---|---|
| **Canopy** | *LAI* | Leaf area index (m$^2$ m$^{-2}$) | 0.0 to 7.0 | Uniform |
| | *LIDF* | Leaf inclination distribution function (°) | 0 to 90 | Gaussian |
| | *hspot* | Hotspot parameter (m m$^{-1}$) | 0.12 | Fixed |
| Leaf | *N* | Leaf structural parameter in PROSPECT | 1.518 | Fixed |
| | $C_{ab}$ | Chlorophyll a + b content in PROSPECT (µg cm$^{-2}$) | 45.0 to 60.0 | Uniform |
| | $C_{ar}$ | Carotenoid content in PROSPECT (µg cm$^{-2}$) | 8.0 | Fixed |
| | $C_{brown}$ | Brown pigment content (ug/cm$^2$) | 0.20 | Fixed |
| | $C_w$ | Equivalent water thickness in PROSPECT (cm) | 0.05 to 0.30 | Gaussian |
| | $C_m$ | Dry matter content in PROSPECT (g cm$^{-2}$) | 0.002 to 0.012 | Gaussian |
| Soil and sky | $p_{soil}$ | Soil reflectance assumed to be Lambertian (1) or not (0) | 0–1 | Gaussian |
| | *skyl* | Ratio of diffuse to total incident radiation | Calculated by $\theta_s$ | Fixed |
| Sun sensor | $\theta_s$ | Solar zenith angle (°) | 29 | Fixed |
| | $\theta_v$ | Viewing zenith angle (°) | 0 | Fixed |
| | $\varphi_{sv}$ | Relative azimuth angle (°) | 0 | Fixed |

Note: The symbol "-" represents a one-set value.

## 3. Results and Analysis

### 3.1. Corn Plant Height Estimation Results

The nDSM was used to estimate the corn plant height for two breeding fields in the Shunyi breeding trial (fields No. 12 and No. 14). These small plots were used for a hybrid trial to test five new corn cultivars. Figure 4b shows the calculated DSM within these plots using the SfM photogrammetry method, Figure 4c shows the interpolated DTM using the bare ground points, and Figure 4d shows the nDSM calculated by subtracting the DTM from the DSM. Figure 4c shows that there were three high microreliefs of bare earth in the northern and southern parts of field No. 12 and in the southwestern part of field No. 14. These three locations showed higher elevations in the DSM (Figure 4b). The nDSM (Figure 4d) accounted for these factors to show the true height of the corn plants. The corn cultivars in the fifth plot from the left within field No. 12 were taller than the other cultivars, and the corn cultivars in the northern part of field No. 14 were taller than those in the southern part. The estimated corn plant heights were compared to the corresponding heights measured in situ. There were 28 corn plants that were randomly measured for their plant heights during the fieldwork for a height estimation accuracy assessment, and the measured heights were compared to the estimated heights (Figure 5). The locations of the measured heights were positioned using a Huace i80 real-time kinematic (RTK) GPS receiver (Huace Ltd., Shanghai, China). The relationship between the measured and estimated heights was strong and significant ($R^2 = 0.7833$, $p < 0.001$) with reasonable accuracy (RMSE = 0.1677).

To test our ability to detect height differences between the different corn cultivars, the plant heights and the coefficient of variation (CV) of all corn cultivars combined were measured in fields No. 12 and 14 in the Shunyi breeding trial. Figure 6a shows a map of the corn plant heights in two fields. The corn plant height for the cultivars in the fifth plot from the left within field No. 12 were greater than those

in the eastern and western parts. The corn plants with the lowest average height were in plot 6-12, with an average height of 0.46 m. The corn plants with the greatest average height were in plot 1-2, with an average height of 2.20 m. In field No. 14, the corn plants in the north were taller than those in the south. Within this area, the corn plants in plot 13-21 were the shortest, with an average height of 0.78 m, and the tallest corn plants were in plot 2-4, with an average height of 2.09 m. These results agreed with the spatial variability shown in Figure 4d.

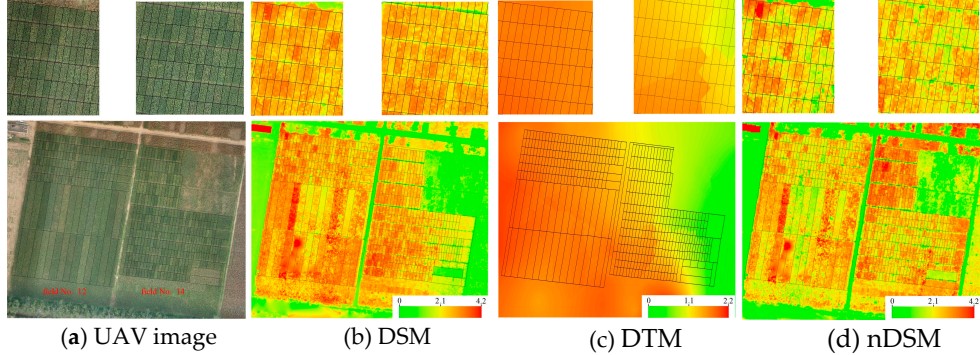

(**a**) UAV image　　　　　(**b**) DSM　　　　　(**c**) DTM　　　　　(**d**) nDSM

**Figure 4.** Examples of the images used to estimate corn plant height: (**a**) UAV orthophoto image mosaic, (**b**) digital surface model (DSM), (**c**) digital terrain model (DTM), and (**d**) normalized DTM in the Shunyi breeding trial. Note: the legend is labeled using the heights above the minimum elevation in the study area.

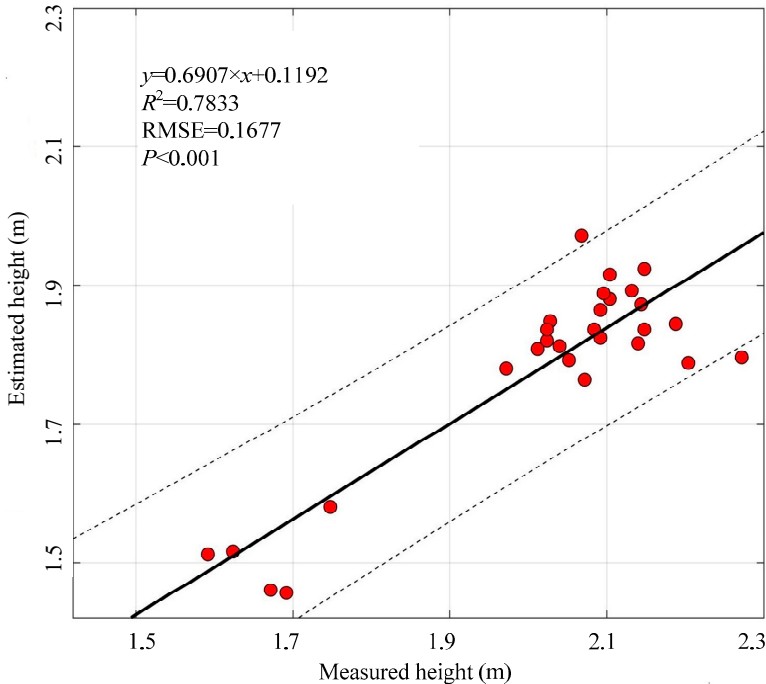

**Figure 5.** Comparison between the heights measured in situ and the estimated heights. The broken lines represent the 95% confidence interval.

The CV of corn plant height represents the consistency of growth by the different corn cultivars. A small CV indicates a high consistency of height. We chose a population standard of CV = 3% and a 95% probability of acceptance to identify corn height cultivars with consistent growth. Figure 6b shows the spatial variation of CVs for the different cultivars. The CV was smallest for the corn cultivars in the western and southeastern parts of field No. 12. The smallest CV occurred in plot 7-3, with a value of 0.09, and the largest CV occurred in plot 6-12, with a value of 1.02. The CVs were more variable in

field No. 14 due to the different sizes of the corn plants. The smallest CV was 0.05, in plot 7-2, and the largest CV was 0.55, in plot 10-20.

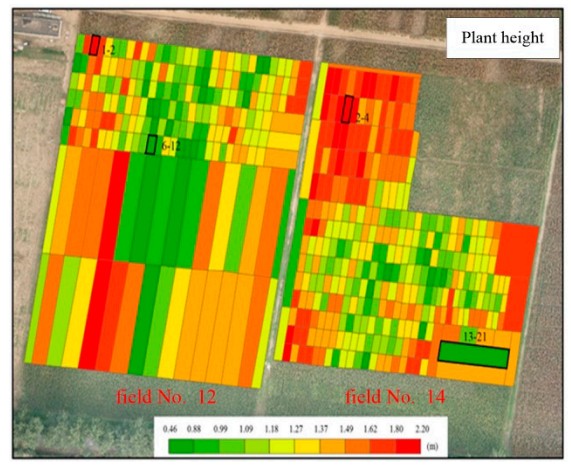
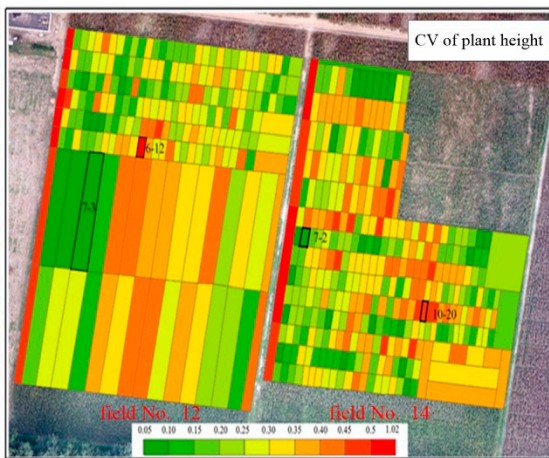

(**a**) Estimated corn plant height　　　　　　　　　(**b**) Calculated CVs of corn plant height

**Figure 6.** Mapping of corn plant height (**a**) and coefficients of variation (CVs) for different corn varieties (**b**) in the Shunyi breeding trial.

## 3.2. Estimated Lodging Area

Because lodging resistance is an important characteristic for corn breeding, there were two methods used to estimate the lodging area. First, the GLCM texture analysis method was used to identify lodging areas using UAV RGB images, as described in Section 2.2.2. Figure 7a includes preprocessed UAV RGB images of field No. 4, field No. 6, and field No. 8: the labeled light green areas are lodging areas, and the labeled dark green areas are nonlodging areas. Figure 7b shows the generated ASM texture feature, and Figure 7c shows the estimated lodging area based on this textural feature. Second, the nDSM was used to detect lodging areas: Figure 7d shows the nDSM, Figure 7e shows the resulting lodging areas, and Figure 7f shows the spatial variation within the lodging area.

To validate the estimation results of these two methods, an accuracy assessment was done by comparing the estimated lodging area and the in situ-measured lodging area for all corn breeding plots in fields No. 4, No. 6, and No. 8. Table 3 is a comparison of the total lodging areas in fields No. 4, No. 6, and No. 8, which revealed that the measured lodging areas agreed reasonably well with the estimates produced by both methods (Table 3). Specially, the accuracy of the nDSM method was much more accurate than that of the textural method. Particularly, the estimation error using the nDSM in fields No. 4 and 6 was 0.85% and 0.97%; however, the error using the ASM in these two fields was 10.0% and 16.2%. Its estimation errors were roughly an order of magnitude smaller than the errors based on the texture analysis.

To compare the lodging resistance of different corn varieties, the proportion of the lodged corn plants in each plot and its CV were calculated. Figure 7e shows the spatial distribution of the percentage of lodging and reveals serious lodging in field No. 4, where there were relatively few lodged corn plants. The lodging was most serious in plots 1-2 and 3-6, where lodging affected 100% of the plants. The least serious lodging was in plot 7-4, with a value of 2.5%. This indicates that the corn variety in plot 7-4 showed good lodging resistance. The lodging was not serious in field No. 6, and only some plots in the east showed significant lodging, such as plots 6-10 and 8-10. The corn variety with the most variable lodging was in field No. 8, with the maximum lodging rate reaching 100% and a minimum lodging rate reaching 0%. Plot 2-1 was a circular area, and lodging was most serious in the center of the plot, reaching 23.3%.

Figure 7f shows the CVs for the lodging area in each plot. The proportion of the plants that experienced lodging roughly equaled the proportion of plants that did not when the CV reached

0.5. When the CV is 0, this indicates that either all corn plants lodged or none of the plants lodged. Figure 7f shows that the maximum CV (0.5) occurred in the northern part of field No. 4, whereas the maximum CV occurred in the middle and western parts of field No. 6, and the CV in field No. 8 was highly variable.

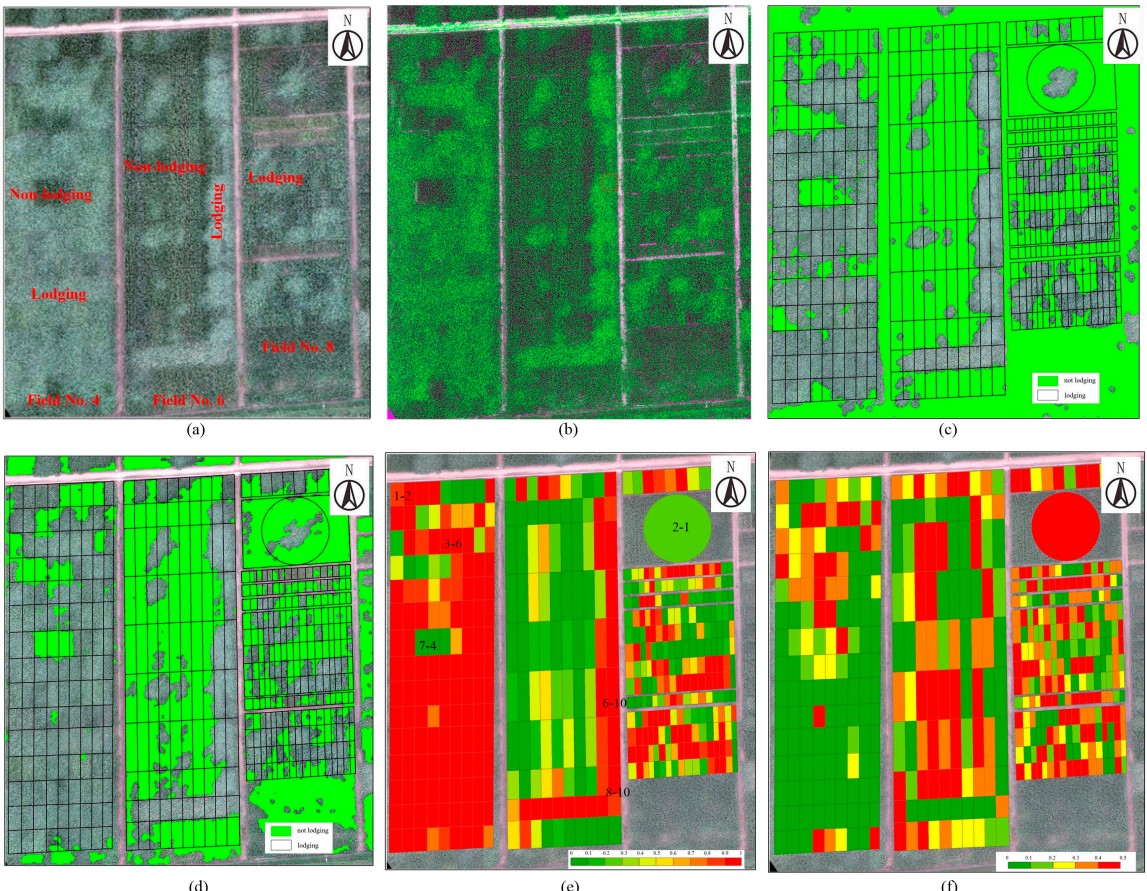

**Figure 7.** Identification of lodging areas based on texture analysis and the nDSM. (**a**) A preprocessed RGB UAV image; (**b**) the spatial distribution based on the ASM texture feature; (**c**) the estimated lodging area based on the ASM texture feature; (**d**) the estimated lodging area using the nDSM; (**e**) the proportion of the lodging area in every plot; and (**f**) the coefficient of variation (CV) of the lodging area in every plot.

**Table 3.** The measured and estimated lodging areas and the resulting estimation errors using the nDSM method and the ASM textural feature.

| Parameters | Field No. | | |
|:---:|:---:|:---:|:---:|
| | **4** | **6** | **8** |
| Measured lodging area (m$^2$) | 6409 | 2993 | 2503 |
| Estimated area using nDSM (m$^2$) | 6464 | 3022 | 2372 |
| Estimation error using nDSM (%) | 0.85 | 0.97 | 5.23 |
| Estimated area using ASM (m$^2$) | 5768 | 2508 | 2203 |
| Estimation error using ASM (%) | 10.0 | 16.2 | 12.0 |

### 3.3. Retrieved LAI Using PROSAIL Model

The corn canopy LAI was retrieved using the parameterized PROSAIL model inputs in Table 2 based on our previous work [39]. According to the range and value of input variables (Table 2), the program identifies a remote sensing image pixel by pixel through a cost function to find the

best fit/match between the observed pixel reflectance and the simulated reflectance in the look-up table. Figure 8 shows the retrieved LAI on 15 July (stem elongation stage) (Figure 8a), 26 July (the flare opening stage) (Figure 8b), 11 August (the heading stage) (Figure 8c), and 4 September (the milking stage) (Figure 8d). In Figure 8, the three horizontal rows are the three repetitions of different fertilization amounts for each column, indicating each treatment with different corn varieties and planting densities. For example, the field labeled 1-7 in Figure 8a had treatment with corn varieties of Xianyu 335 with 82,500 plants/ha. Due to experimental secrets, the fertilization amount cannot be listed here. Figure 8a,b show that the retrieved LAIs for treatments of *-7 (1-7, 2-7, and 3-7) and *-23 (1-23, 2-23, and 3-23) were higher than for other treatments at the stem elongation stage (Figure 8a) and the flare opening stage (Figure 8b). Here, *-23 was treatment with corn varieties of Xianyu 335 with 12,000 plants/ha. This revealed that the corn variety of Xianyu 335 grew quickly with other corn varieties. At the heading stage (Figure 8c), the LAI of most treatments reached 6.00–7.00, except for the corn varieties that included *-13 (1-13, 2-13, and 3-13), *-15 (1-15, 2-15, and 3-15), and *-24 (1-24, 2-24, and 3-24), which were 3.00–4.00. Figure 8d shows that there were two treatments, including *-7 (1-7, 2-7, and 3-7) and *-23 (1-23, 2-23, and 3-23), that got higher LAIs in the milking stage. Unfortunately, this higher LAI was not a good growth condition, and these areas were lodged corn plants. This result reveals that the lodging resistance of Xianyu 335 was weak.

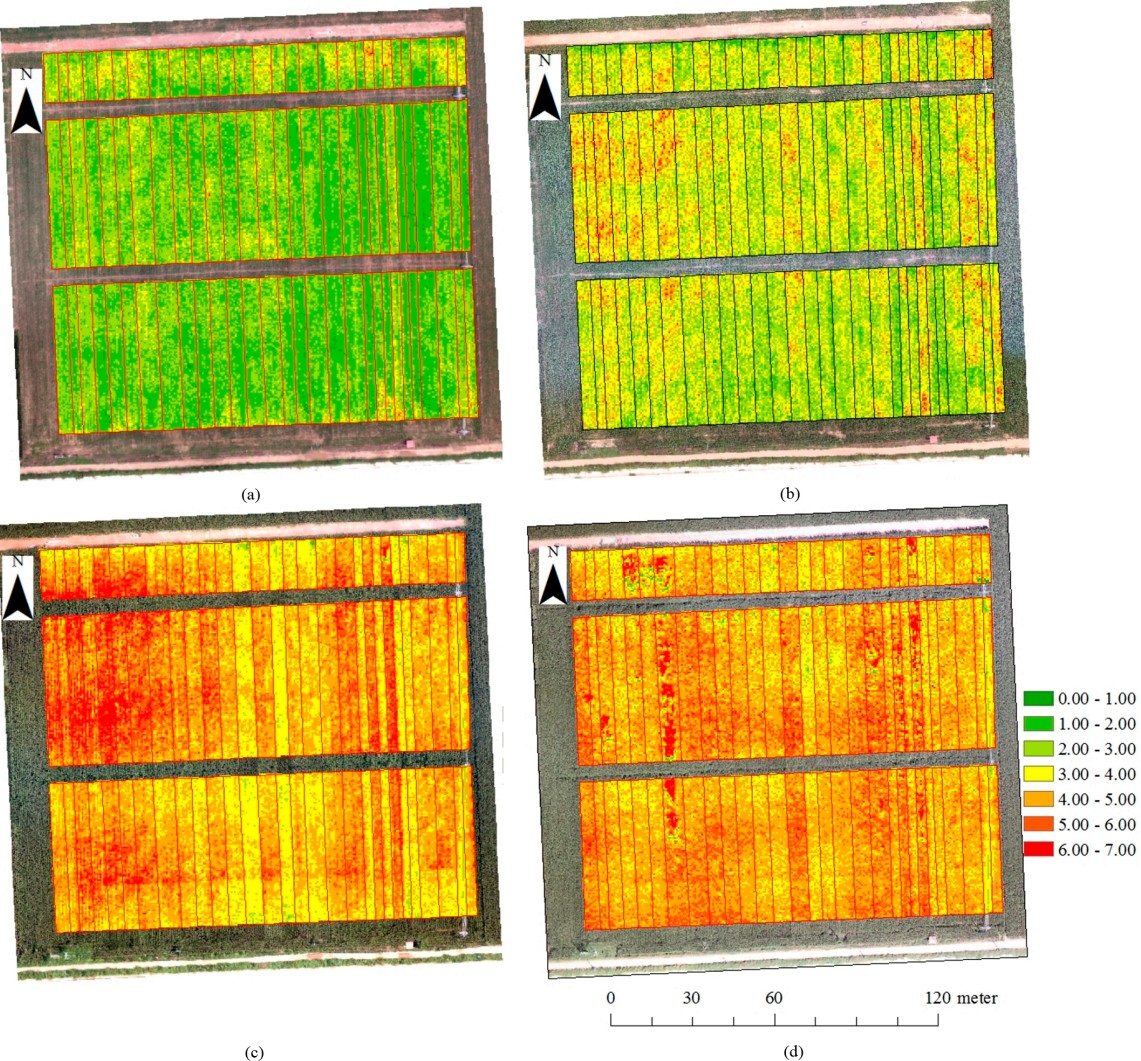

**Figure 8.** The retrieved LAI on (**a**) 15 July (stem elongation stage), (**b**) 26 July (the flare opening stage), (**c**) 11 August (the heading stage), and (**d**) 4 September (the milking stage).

There were 47, 62, 61, and 42 in situ-measured LAIs in the field campaign on 15 July, 26 July, 11 August, and 4 September, respectively. All of the measured data were used to validate the LAI retrieval accuracy. Figure 9 shows the accuracy assessment results of retrieved LAIs using the PROSAIL model compared to in situ-measured LAIs on 15 July (stem elongation stage) (Figure 8a), 26 July (the flare opening stage) (Figure 8b), 11 August (the heading stage) (Figure 8c), and 4 September (the milking stage) (Figure 8d). The correlation coefficients of the retrieved and measured LAI (with $P < 0.01$) on 15 July (Figure 8a), 26 July (Figure 8b), 11 August (Figure 8c), and 4 September (Figure 8d) were 0.6052, 0.7490, 0.7450, and 0.7233, respectively. Comparably speaking, the retrieved accuracy of LAIs in the early growing stage (i.e., 15 July, the stem elongation stage in this study) was lower than in the middle and later growing stages because the corn canopy was not closed and the image pixels were a mixture of the corn canopy and bare soil in the early growing stage.

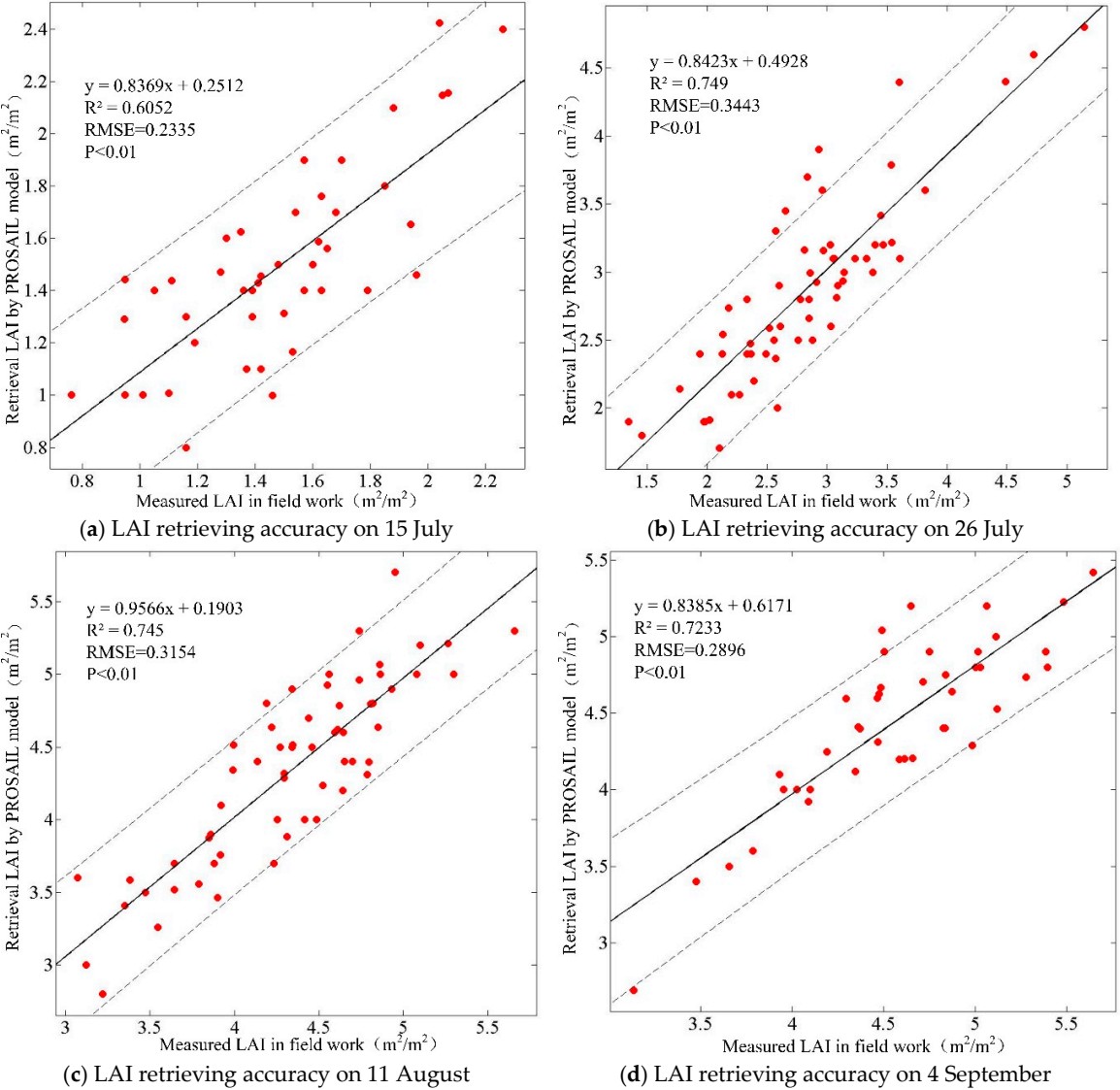

(**a**) LAI retrieving accuracy on 15 July

(**b**) LAI retrieving accuracy on 26 July

(**c**) LAI retrieving accuracy on 11 August

(**d**) LAI retrieving accuracy on 4 September

**Figure 9.** Comparison between the LAIs measured in situ and the retrieved LAIs using the PROSAIL model on (**a**) 15 July (stem elongation stage), (**b**) 26 July (the flare opening stage), (**c**) 11 August (the heading stage), and (**d**) 4 September (the milking stage). The broken lines represent the 95% confidence interval.

## 4. Discussion

Phenotyping is an urgent demand in crop breeding currently. Crop breeders need the geometric traits and biochemical traits of crop plants and canopies to identify the genetic compositions of lines within different crop species. UAV images are effective ways to phenotype nondestructively at a low cost. A methodology for estimating breeding corn plant heights, lodging areas, and canopy LAIs using calibrated UAV images has been proposed in this study. The motivation for this study was to find effective ways to estimate corn plant height differences and their coefficients of variation, lodging proportions and their coefficients of variation, and canopy LAIs. This phenotyping did help to test the agronomic characters of new corn varieties, including lodging resistance, leaf color, etc.

The corn plant heights were estimated using the nDSM, which is the difference between the DSM and DTM. In other words, the nDSM is a height model of land objects above the ground, i.e., the corn plants in this study. The DSM is calculated from stereoscopic UAV images using SfM photogrammetry [43]. The DTM is restored by selecting the bare ground points using the hierarchical moving curve-fitting method [31] and interpolating using ordinary kriging interpolation [44]. The corn plant height estimation results show that this method was effective in estimating regional corn plant heights. Certainly, the height estimation accuracy was low, with $R^2 = 0.7833$ in this study. The accuracy could be improved by more longitudinal overlap and sidelap. Unfortunately, further overlap was limited by the flying plans in the whole study area. The plant heights varied obviously for different corn varieties in the Shunyi corn breeding trial, Beijing, China. The heights of the corn varieties planted in plot 6-12 and plot 13-21 were lower, shorter than 1 m. Comparatively, the corn varieties planted in plot 1-2 and plot 2-4 were a long-stalked variety with an average height taller than 2 m. Based on the estimated corn plant height results, we estimated the CVs of the height changes from this plant to that plant in studied fields No. 12 and No. 14, which were an indicator of corn plant height regularity within the tested corn varieties. Our results show that the corn varieties planted in plot 7-3 and plot 7-2 had the best plant height regularity, with a CV value of 0.09 and 0.05, respectively. The corn variety planted in plot 6-12 had the lowest plant height regularity, with a CV value of 1.02. Based on huge previous corn height estimation research, we took advantage of the estimated corn plant heights of 215 plots in the Shunyi corn breeding trial. This height estimation was done to find the corn varieties with optimal plant heights and lodging resistance. Moreover, the CVs of the plant heights were analyzed to depict the corn canopy uniformity, which was used to validate the stability of a corn variety.

There were two kinds of methods used in the lodging area estimation in this study, the GLCM texture analysis method and the nDSM calculating method: the former is an image analysis method [33], and the latter is a photogrammetry method. Our estimation results from the Shunyi corn breeding trial demonstrated that the nDSM method was more accurate than the GLCM texture analysis method, with their smallest estimation error values being 0.85% and 10.0%, respectively. The lodging area estimation results in this study show that the corn varieties planted in plots 1-2, 3-6, 6-10, and 8-10 had the poorest lodging resistance, with a 100% lodging area, under the same meteorological conditions as other corn varieties in the Shunyi corn breeding trial.

There are some studies used vegetation indexes of UAV images as crop growth, crop vigor, or an indicator of crop leaf numbers [45–47]. Unfortunately, vegetation indexes such as the NDVI are easily saturated when crop plants are thick and the canopy is closed [48,49]. Therefore, we used the PROSAIL radiative transfer model to retrieve corn canopy LAI as a corn canopy geometric phenotypic trait based on UAV multispectral images with green, red, red-edge, and near-infrared bands in Baogaofeng Farm, Mazhuang Town, Xinji City, Hebei Province, China. Our results revealed that (1) the PROSAIL model could be used to get high-accuracy LAI retrieval results at all growing stages of corn. The correlation coefficients ($R^2$) between retrieved LAI and in situ-measured LAI were 0.6052, 0.7490, 0.7450, and 0.7233 on 15 July, 26 July, 11 August, and 4 September, respectively. (2) For the tested corn varieties in Baogaofeng Farm, the corn variety of Xianyu 335 had rapid growth in the early and middle growing season. However, the lodging resistance of Xianyu 335 was weak.

This study demonstrates the ability to estimate corn plant height, lodging area, and canopy LAI using UAV RGB images and multispectral images. These attempts benefitted the phenotyping of corn breeding. However, there was one limitation that are worth noting. It is that there were only three phenotypic parameters that were studied. This is just a glimpse into phenotyping using a UAV technique. There are more phenotypic parameters that should and can be estimated from UAV images [4,5] to mine the big potential of UAV techniques and promote modern corn breeding development.

## 5. Conclusions

In this study, the potential of phenotyping to support corn breeding based on an analysis of UAV images in visible and multispectral bands was investigated. Structural characteristics (plant height, canopy LAI) and gene expression (lodging area) were estimated using visible bands. The following was found:

1. The stereoscopic and photogrammetric methods showed promise for calculating corn plant heights from UAV images, with $R^2$ = 0.7833 and RMSE = 0.1677. This approach let us distinguish corn cultivars with different heights over a wide range of heights and estimate the CV of heights for each variety;

2. The nDSM method provided a more accurate estimate of the corn lodging area than the GLCM textural features method sis, with errors <6% and >10%, respectively. In addition, we were able to estimate the percentage of corn lodging in each plot and its CV, thereby identifying cultivars with potentially higher lodging resistance;

3. The PROSAIL radiant transfer model with multispectral inputs offered more accurate and robust characterization of corn phenotypes in terms of corn canopy LAI.

These comparisons and analyses on corn plant heights, lodging areas, and canopy LAIs were done to find the optimal method for phenotyping with high-throughput nondestructively. In our future work, we will investigate the potential of UAVs in phenotyping by examining this technique's ability to detect other phenotypic traits, such as vegetation cover, emergence dates, signs of abiotic or biotic stress, and leaf nitrogen concentrations. In addition, we will examine the phenotypic responses of corn plants at different planting densities and under different fertilization regimes. Finally, we will test the association between field UAV results and laboratory determinations of genomic characteristics for a range of corn cultivars.

**Author Contributions:** This work was a cooperation of our research team, and the contributions were as follows: conceptualization, W.S. and Z.L.; methodology, J.H.; investigation, D.B.; software, M.Z. and W.W.; writing—original draft preparation, W.S.; writing—review and editing, Z.L.; validation, H.G.; visualization, J.W.; project administration, W.S.; funding acquisition, NSFC.

**Funding:** This study was funded by the National Key Research and Development Program of China (No.2017YFD0300903), the National Natural Science Foundation of China under the projects "Growth process monitoring of corn by combining time series spectral remote sensing images and terrestrial laser scanning data" (41671433) and "Estimating the leaf area index of corn in whole growth period using terrestrial LiDAR data" (41371327), the Chinese Universities Scientific Fund ("Retrieval of biomass for summer corn after canopy is closed using remote sensing image" (No. 2019TC138) and "Monitoring the quantity and quality using remote sensing and intelligence analysis of total factor for cropland" (No. 2019TC117)), and the Science and Technology Facilities Council of the UK–Newton Agri-tech Programme (Project No. ST/N006798/1).

**Conflicts of Interest:** The authors declare no conflict of interest.

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
