# Peer review of "Phenotyping of Corn Plants Using Unmanned Aerial Vehicle (UAV) Images"

_remotesensing, doi:10.3390/rs11172021_

Round 1
Reviewer 1 Report
@page { margin: 0.79in } p { margin-bottom: 0.1in; line-height: 120% } a:link { so-language: zxx }This manuscript presents the use of UAV images to estimate plant height, lodging area, and canopy LAI for corn plants. The authors did experiments on two corn breeding trials. The work demonstrates that, UGA technique can provide a promising method for plant phenotyping. It’ll be nice to have more details in some parts.
1. What the red points and blue plus signs mean in Figure 2?
2. The “2.1 Study area and field campaign” lacks details about the field experiments. For example, the number of plots/cultivars, plant densities, ect.
3. Section “2.3 UAV image processing” and “2.4 phenotyping methodilogy” lack details to repeat the work.
a) It is not clear which steps were completed using Pix4D software, and which steps were completed using custom code.
b) Line 186, 209, 213: How to create the DSM, DTM? Was SfM used to create point cloud or DSM?
c) Line 218: There should be many plants in the model. How to estimate the plant height for individual plants? Was there only one local maximum value for each plant?
d) Line 234: Which channel was used to compute the ASM feature? How to extract the lodging area from the image of ASM? By threshold or other image processing methods?
e) Line 236: How to classify lodged area and un-lodged areas from nDSM?
4. Line 278: How did the authors choose these 28 plants?
5. Line 275, 295: I think the plants in the middle of field were shorter than those in the eastern and western.
6. Line 326-329, Table 2: Three samples for validation? Is that enough?
7. There are not field No.6 and No.8 shown in Figure 7e and 7f.
Author Response
Dear Reviewer:
Thank you for your letter and for the reviewers’ comments concerning our manuscript entitled “Phenotyping of corn plants using unmanned aerial vehicle (UAV) images” (Ref: remotesensing-553815). Your comments are all valuable and very helpful for revising and improving our paper, as well as the important guiding significance to our researches. We have improved the methods, such as where the steps for DSM, DTM, nDSM, plant heights estimation, ASM feature, lodging area estimation. Moreover, the English has been checked. In the rest of the letter, we have provided specific responses to each review comment.
We are grateful for the review comments, and we hope that our responses and the resulting changes in the manuscript will be satisfactory. However, we will be happy to work with the editor and reviewers to resolve any remaining issues. Sincerely,
Wei Su, Mingzheng Zhang, Dahong Bian, Zhe Liu, Jianxi Huang, Wei Wang, Jiayu Wu, Hao Guo
Reply to comments:
1. What the red points and blue plus signs mean in Figure 2?
Reply: red points are the location of pictures taken by UAV, and the blue plus signs are the location of ground GPSs. This sentence has been added as note from Line 183 to Line 184.
2. The “2.1 Study area and field campaign” lacks details about the field experiments. For example, the number of plots/cultivars, plant densities, ect.
Reply: the details of corn breeding experiments were added from Line 136 to Line 142 and Line 153 to Line 156. For the reason of corn breeding secret, the corn cultivars in Shunyi were named using number codes.
3. Section “2.3 UAV image processing” and “2.4 phenotyping methodilogy” lack details to repeat the work.
a) It is not clear which steps were completed using Pix4D software, and which steps were completed using custom code.
b) Line 186, 209, 213: How to create the DSM, DTM? Was SfM used to create point cloud or DSM?
c) Line 218: There should be many plants in the model. How to estimate the plant height for individual plants? Was there only one local maximum value for each plant?
d) Line 234: Which channel was used to compute the ASM feature? How to extract the lodging area from the image of ASM? By threshold or other image processing methods?
e) Line 236: How to classify lodged area and un-lodged areas from nDSM?
Reply: a) ① the image mosaic, geometric correction, creation of ortho-mosaics, radiometric calibration, DSM generation were done using the Pix4Dmapper software, which were added from Line 186 to Line 187. ② And the selecting bare ground points, interpolation and calculation of difference between DSM and DTM were completed using our custom code, which were added from Line 238 to Line 239, Line 247 to Line 249.
③ GLCM textures computation were done in the ENVI software (version 5.3), which were added from Line 272.
④The code of forward PROSAIL Python Bindings was downloaded from https://pypi.org/project/prosail/, which were added from Line 310 to Line 311.
⑤ The codes of backward LAI retrieval using PROSAIL model and improvement of corn leaf inclination distribution function were our custom code, which were added from Line 316 to Line 317.
b) the Line 186 comes early, which result in confusion. So we delete it now. Line 209 and 213 are redundant and unclearly. So we re-organized as sentence from Line 238 to Line 239, Line 243 to Line 247 . The actual meaning is: the 3D points were generated from stereo UAV images using the SfM method, and the generated 3D points were used to created a DSM in Pix4D software.
c) yes, exactly. There were many corm plants in the nDSM model. Because the DSM, DTM and nDSM were all store using raster pixel format, and the height of local lowest/highest points was calculated as the grid value. So the height of every pixel were analyzed in this study, not the plant height for individual plants. The sentence in Line 218 is confused too. So we update this description using “pixel by pixel” in Line 247.
d) Line 234: Which channel was used to compute the ASM feature? How to extract the lodging area from the image of ASM? By threshold or other image processing methods?
The blue, green and red bands of Sony DSC QX100 digital camera pictures were used to extracted the ASM textural features for estimating lodging area. And the Figure 7 (b) was the bands combination result of three textural bands. The lodging area was extracted using supervised classification method of SVM. The description was added from Line 273 to Line 275.
e) Line 236: the height difference was determined by the threshold which were captured by statistics of lodging and non-lodging areas in study area. This has been added from Line 277 to Line 278.
4. Line 278: How did the authors choose these 28 plants?
Reply: these 28 plants were selected randomly in field No. 12 and No. 14, and 5 of them were about 1.7m and 23 of them were about 2.0m to 2.3m. And the locations of measured height were positioning using a Huace i80 real-time kinematic (RTK) GPS receiver.
5. Line 275, 295: I think the plants in the middle of field were shorter than those in the eastern and western.
Reply: yes, indeed. What we want to depict was the fifth plot from left was taller than other plots. So we revised this description in Line 342 and Line 365 now.
6. Line 326-329, Table 2: Three samples for validation? Is that enough?
Reply: these three results in Table 2 are the comparation of total lodging areas in field No, 4, No. 6 and No. 8, and there were many plots in every filed. For example, there were 150 plots in field No, 4. Therefore, this is the comparation of sum of estimated lodging area.
7. There are not field No.6 and No.8 shown in Figure 7e and 7f.
Reply: I am very sorry for this neglect, so we have added the labels for all three fields, field No, 4, No. 6 and No. 8, in Figure 7.

Reviewer 2 Report
Dear Auth
The authors have used UAV based image to study the phenotyping feature of corn crop to improve and produce accurate result for corn breeding program. Different measurement were utilized to estimate geometric traits: plant height, canopy leaf area index (LAI) and 15 lodging-resistant trait (lodging area). The results are promising, and I hope the authors keep continue doing this kind of research to improve breeding program for any economic crop. The manuscript is well organized.
Here are some minor issues need to fix:
Line 16: delete “we” use passive voice.
Line 24: UAV can be used for crops not only corn, so change this sentence “UAV appears to provide a promising method to support phenotyping
for corn breeding.”
Line 39: Change “speed” to rapid.
Line 42: Do you mean the data loss only for cloud cover case, find other reasons please. “it has potential on improving the identification of desirable traits and reducing the risk of data loss due to cloud cover”
Line 48-52: you have to focus on which crop and which method had been used, not general information “extensively reviewed the advances that have been achieved in field-based phenotyping using UAVs. They summarized the key traits to support crop breeding, such as the ability to detect the crop’s geometric traits, phenotype-related spectral indices, crop physiological traits, crop abiotic and biotic stress symptoms, and nutrient status, and to allow prediction of the potential crop yield.”
Line 58-63: 58: It is too long making it shorter “Shi et al. [1] investigated the ability of remote sensing using a fixed-wing Anaconda UAV (Ready Made RC, Lewis Center, Ohio) and an X88 rotary-wing UAV for automate collection of plant traits. They found that the correlations between the average normalized difference vegetation index (NDVI) and the calculated digital surface model (DSM) created from steric UAV images could be combined with manually measured heights of corn and sorghum to estimate the height trait in the whole growing season”
Line 93: I am not sure why you mentioned references in beginning and then you added it in last of the sentence, add year to the reference “Duan et al.”
Line 95: You have to choose to add your reference in the beginning or in the last paragraph. Add year to the reference “Shi et al.”
Line 98: Add year to the reference “Roosjen et al.”
Line 100: Add year to the reference” Xu et al.”
Line 110: delete “We”. Use passive voice.
Line 113: Delete “We”.
Line 147: Delete “we”
Line 160: Use spectroradiometric instead of “spectrometer”.
Line 162: delete “We”.
Line 165: delete “We”.
Line 173: Why did you use 6 m/s?
175: “And” should not come in beginning of the new sentence.
Line 191: Delete “And”
Line 193: Delete “and”.
Line 197: It is better to replace it with color picture.
Line 316: Replace “We” with passive voice.
Line 318: Change “we”. Try to fix any “we” with passive voice.
Line 389: add (a,b,c,d) to your pictures.
Good luck.

Author Response
Dear Reviewer:
Thank you for your letter and for the reviewers’ comments concerning our manuscript entitled “Phenotyping of corn plants using unmanned aerial vehicle (UAV) images” (Ref: remotesensing-553815). Your comments are all valuable and very helpful for revising and improving our paper, as well as the important guiding significance to our researches. We have fixed the issues and checked the English. In the rest of the letter, we have provided specific responses to each review comment.
We are grateful for the review comments, and we hope that our responses and the resulting changes in the manuscript will be satisfactory. However, we will be happy to work with the editor and reviewers to resolve any remaining issues. Sincerely,Wei Su, Mingzheng Zhang, Dahong Bian, Zhe Liu, Jianxi Huang, Wei Wang, Jiayu Wu, Hao Guo
Reply to comments:
1. Line 16: delete “we” use passive voice.
Reply: yes, and the “We found that” had been revised into “It was found that”.
2. Line 24: UAV can be used for crops not only corn, so change this sentence “UAV appears to provide a promising method to support phenotyping for corn breeding.”
Reply: yes, indeed. So we change this sentence to “UAV appears to provide a promising method to support phenotyping for crop breeding, and the phenotyping of corn breeding in this study validate this application”.
3. Line 39: Change “speed” to rapid.
Reply: the “speed” has been changed into rapid now.
4. Line 42: Do you mean the data loss only for cloud cover case, find other reasons please. “it has potential on improving the identification of desirable traits and reducing the risk of data loss due to cloud cover”
Reply: yes. We have extended to “cloud/raining/smog cover and limitation resulting from the long revisit periods of satellites” in Line 42 now.
5. Line 48-52: you have to focus on which crop and which method had been used, not general information “extensively reviewed the advances that have been achieved in field-based phenotyping using UAVs. They summarized the key traits to support crop breeding, such as the ability to detect the crop’s geometric traits, phenotype-related spectral indices, crop physiological traits, crop abiotic and biotic stress symptoms, and nutrient status, and to allow prediction of the potential crop yield.”
Reply: yes, so we compressed these two sentences and added the already works about corn plant height estimation using UAV based LIDAR points data, corn lodging estimation using the spectral and textural difference, soybean canopy LAI estimation using spectral vegetation indexes, specifically, from Line 49 to Line 58.
6. Line 58-63: 58: It is too long making it shorter “Shi et al. [1] investigated the ability of remote sensing using a fixed-wing Anaconda UAV (Ready Made RC, Lewis Center, Ohio) and an X88 rotary-wing UAV for automate collection of plant traits. They found that the correlations between the average normalized difference vegetation index (NDVI) and the calculated digital surface model (DSM) created from steric UAV images could be combined with manually measured heights of corn and sorghum to estimate the height trait in the whole growing season”
Reply: yes. So we have compressed these two sentences into three lines form Line 64 to Line 67.
7. Line 93: I am not sure why you mentioned references in beginning and then you added it in last of the sentence, add year to the reference “Duan et al.”
Reply: it has been changed into Duan et al. [12] now.
8. Line 95: You have to choose to add your reference in the beginning or in the last paragraph. Add year to the reference “Shi et al.”
Reply: it has been changed into Shi et al. [1, 6] now.
9. Line 98: Add year to the reference “Roosjen et al.”
Reply: it has been changed into Roosjen et al. [20] now.
10. Line 100: Add year to the reference” Xu et al.”
Reply: it has been changed into Xu et al. [21] now.
11. Line 110: delete “We”. Use passive voice.
Reply: it has been changed into “Determining if SfM photogrammetry method can be used to……” now.
12. Line 113: Delete “We”. Line 147: Delete “we”
Reply: it has been changed into “Comparing the corn plants height estimation accuracy using …………. And the corn plant heights were calculated from the…….” And “Exploring the potential of LAI retrieving using our improved PROSAIL (iPROSAIL) radiative transfer model based on UAV images.” now.
13. Line 160: Use spectroradiometric instead of “spectrometer”.
Reply: the “spectrometer” has been changed into “spectroradiometric”.
14. Line 162: delete “We”.
Reply: this sentence has been changed into passive voice “The corn plant heights were measured using a meter stick from the ground to the tip of the plant”.
15. Line 165: delete “We”.
Reply: this sentence has been changed into passive voice “The DJI S1000+ UAV (DJI-Innovations Co., Ltd., Shenzhen, China; Figure 2a) was used to……”
16. Line 173: Why did you use 6 m/s? 175: “And” should not come in beginning of the new sentence.
Reply: the 6 m/s is determined by the timing interval of UAV taking photo and the flight altitude. The “And” has been deleted now.
17. Line 191: Delete “And”. Line 193: Delete “and”.
Reply: these “And” and “and” have been deleted now.
18. Line 197: It is better to replace it with color picture.
Reply: On the one hand, the radiation calibration board was black and white. On the other hand, there were only green, red, red-edge and NIR bands for Sequoia images. So they couldn’t be layer stacked as true color picture. Thus, we used the four bands picture individually for depicting the differences of radiation calibration between different bands here.
19. Line 316: Replace “We” with passive voice.
Reply: yes, it has been changed into passive voice “the GLCM texture analysis method was used to identify lodging area using the UAV RGB images” now.
20. Line 318: Change “we”. Try to fix any “we” with passive voice.
Reply: this sentence has been change into passive voice “the nDSM was used to detect”. In addition, the “we” in Line 13, Line15-16, Line 112, Line 114, Line117 , Line 119, Line121-122 , Line150-151 , L ine 171, Line 173, Line 192, Line 210-211, Line 218, Line 243-246, Line254 , Line270 , Line 273, Line 344, Line 361-362, Line 384 etc. have been changed too.
21. Line 389: add (a,b,c,d) to your pictures.
Reply: yes, the (a,b,c,d) have been added into Figure 8 now.

Reviewer 3 Report
In my opinion, the overall paper is well written to read and understand. The topic seems actual, since high quality of data is required for further processing steps, such as classification or variable retrievals. The results provided in this paper are quite interesting, and the paper is technically correct with proper references. The paper seems well structured und is written in an understandable way. The English language of the manuscript should be improved. Some place there is too confusing symbols which can be clarify more precisely. In my opinion, the paper can only publish after addressing following minor comments,
1. The structure of the manuscript is confusing. I suggest the authors to add a schematic view of the used methodology in order to clarify the content.
2. I suggest the authors to explain more about the image pre-processing. Did you apply atmospheric correction? What are the pre-processing that you have applied to the images?
3. All mathematical terms should be properly explain on it’s first use, e.g. eqn. (1) and eqn. (2)
4. In line 251, I think that there is a problem with the notation. It seems that the value of parameters was chosen arbitrarily, Please elaborate more on your decision to use all parameters.
5. How is PROSAIL based radiative transfer model used for yield estimation?
6. How you chose the values for “PROSAIL variables” in the proposed model?
7. On what basis you minimized the mean absolute percentage error (RMSE) of prediction for yield estimation accuracy?
8. “The iPROSAIL model all can be used to get high accuracy LAI retrieved result at the whole growing stages of corn”. (line no. 452) How much is the accuracy, any value/percentage?
9. How you define the coefficient of variation (CV) to simplify the model framework?
10. What threshold values of overall accuracy you considered in RMSE statistic?
11. What is the observation characteristic of using PROSAIL model? Please define properly in the text.
12. Define the formulation of models used in Table 3.
13. Present an explicit and clear algorithmic steps used in this study data simulation.
14. Use the high resolution image for Figure-1 and Figure-4.
15. There is a need of couple of more proper reference to support this study.
Also, the paper should be proofread for sentences flow, English grammar correction, and spelling mistakes.
Author Response
Dear Reviewer:
Thank you for your letter and for the reviewers’ comments concerning our manuscript entitled “Phenotyping of corn plants using unmanned aerial vehicle (UAV) images” (Ref: remotesensing-553815). Your comments are all valuable and very helpful for revising and improving our paper, as well as the important guiding significance to our researches. We have re-organized the Section 2 of “Materials and methods”. Moreover, the English has been checked. In the rest of the letter, we have provided specific responses to each review comment.
We are grateful for the review comments, and we hope that our responses and the resulting changes in the manuscript will be satisfactory. However, we will be happy to work with the editor and reviewers to resolve any remaining issues. Sincerely,Wei Su, Mingzheng Zhang, Dahong Bian, Zhe Liu, Jianxi Huang, Wei Wang, Jiayu Wu, Hao Guo
Reply to comments:
1. The structure of the manuscript is confusing. I suggest the authors to add a schematic view of the used methodology in order to clarify the content.
Reply: yes, this confusion is resulting from the Section 2, and we have re- arranged Section 2 now using "2.1 Materials" and "2.2. Methods". Secondly, a schematic view of the used methodology was added from Line 228 to Line 234. Thirdly, description about inputs of PROSAIL model were added from Line 297 to Line 305, and the Table 3 about the ranges and distributions of the iPROSAIL model was added.
2. I suggest the authors to explain more about the image pre-processing. Did you apply atmospheric correction? What are the pre-processing that you have applied to the images?
Reply: the image mosaic, geometric correction and creation of ortho-mosaics were done using the Pix4Dmapper software, which were depicted from Line 186 to Line 187. And the atmospheric correction was done using the pictures taken from test pattern. The Sequoia drone was placed over the test pattern hold by operator and used to take photo for capturing the entire test pattern and calibrating the Sequoia. And these pictures of test pattern taken were added to Pix4Dmapper software for radiometric calibration during the image mosaic process. These depictions were added from Line 192 to Line 199.
3. All mathematical terms should be properly explain on it’s first use, e.g. eqn. (1) and eqn. (2)
Reply: the introduction about N, Cab, Car, Cw, Cm, LAI, LIDF, hspot, θv, θs, φsv in eqn. (1) was added from Line 284 to Line 286, Line 288 to Line 290, Line 298 to Line 305. The introduction about RMSE were added from Line 324.
4. In line 251, I think that there is a problem with the notation. It seems that the value of parameters was chosen arbitrarily, Please elaborate more on your decision to use all parameters.
Reply: thank you for pointing out this problem. We have updated all the inputs in Table 3 and summarized into four kinds as followed Line 297 to Line299. The LAI comes from the measured LAI value in field work using LAI-2200 Plant Canopy Analyzer, the Cab is measured with a SPAD-502 chlorophyll meter, Cw is tied to the difference of fresh leaf weight and dry leaf weight (Cw =(Cfresh leaf- Cdry leaf)/LAI. All of them have been depicted from Line 299 to Line 305.
5. How is PROSAIL based radiative transfer model used for yield estimation?
Reply: Generally speaking, the PROSAIL model can’t be used for yield estimation directly. This model can be used to retrieve crop canopy parameters, such as LAI, Cab etc., firstly. In the next step, these crop canopy parameters would be used to estimate yield. So this issue doesn’t be mentioned in this study.
6. How you chose the values for “PROSAIL variables” in the proposed model?
Reply: The LAI comes from the measured LAI value in field work using LAI-2200 Plant Canopy Analyzer, the Cab is measured with a SPAD-502 chlorophyll meter, Cw is tied to the difference of fresh leaf weight and dry leaf weight (Cw =(Cfresh leaf- Cdry leaf)/LAI. All of them have been depicted from Line 299 to Line 305.
7. On what basis you minimized the mean absolute percentage error (RMSE) of prediction for yield estimation accuracy?
Reply: this minimization is used to find the best fit between simulated reflectance of all bands and the UAV image reflectance of all bands, aiming at retrieving LAI in this study.
8. “The iPROSAIL model all can be used to get high accuracy LAI retrieved result at the whole growing stages of corn”. (line no. 452) How much is the accuracy, any value/percentage?
Reply: The descriptions about accuracy assessment were added from Line 518 to Line 519. And the correlation coefficients (R2) between retrieved LAI and in-situ measured LAI are 0.6052, 0.7490, 0.7450 and 0.7233 on15th July, 26th July, 11th August, and 4th September, respectively.
9. How you define the coefficient of variation (CV) to simplify the model framework?
Reply: the coefficient of variation (CV) of all corn cultivars in field No. 12 and 14 in the Shunyi breeding trial was used to test the ability to detect height differences among the different corn cultivars. On the one hand, this height differences are needed in corn breeding. On the other hand, the plant heights estimated in this study is the height pixel by pixel. Therefore, there were many heights in every corn breeding plots. The coefficient of variation (CV) characterize the group differences for every corn variety.
10. What threshold values of overall accuracy you considered in RMSE statistic?
Reply: we think RMSE should be lower than 1 for LAI retrieval using remote sensing. Of course, the validation using RMSE value should consider the absolute LAI value. Our previous work, references (Combal et al., 2003; Locherer et al., 2015) all showed the RMSE value less than 1.
11. What is the observation characteristic of using PROSAIL model? Please define properly in the text.
Reply: the sensitivity of PROSAIL model observation characteristics were added from Line 310 to Line 314.
12. Define the formulation of models used in Table 3.
Reply: we have updated Table 3, and added the note about the calculation of skyl.
13. Present an explicit and clear algorithmic steps used in this study data simulation.
Reply: the forward simulation steps of PROSIAL model has been added from Line 315 to Line 324.
14. Use the high resolution image for Figure-1 and Figure-4.
Reply: The Figure-1 and Figure-4 have been placed now.
15. There is a need of couple of more proper reference to support this study.
Reply: some references have been added, such as the references about PROSAIL forward simulation, the reference about computation of GLCM textural features. In addition, the original references 34 to 41 have been updated now.
16. Also, the paper should be proofread for sentences flow, English grammar correction, and spelling mistakes.
Reply: a native English researcher has help us for double checking the English grammar, spelling mistakes etc. For example, “test pattern” has been changed into “radiation calibration board”, all the “we” sentences have been changed into passive voice, “The vegetation indexes method is simple and convenient, however, it needs…” has been changed into “The vegetation indexes method is simple and convenient. However, they need….” etc.

Reviewer 4 Report
The paper discusses the potential of phenotyping suing UAV RGB imagery. The topic is interesting and important for precision agriculture applications.
Although the paper is well organized, it omits some important description about the methods used in the research e.g. GLCM,...and refers reader to find about the methods in references. It would be helpful if some basic information about the methods be included.
The contribution of this paper is not clear. There is a huge literature to estimate and study corn height using SFM and texture analysis. What is the contribution of your research compared with your pervious work and literature?
How many GCPs were used and how was their distribution?
For radiometric calibration, at what height/altitude the pictures were taken from the test pattern?
Line 218: how did you calculate the local maximum height of the nDSM? Using a n*n window? Local maximum can be a noise.
Line 246: “.. for the SAIL model to retrieve LAI using canopy inputs such as LAI, the leaf angle….” . should it be “ such as measured LAI, ..” ?
Captions and description of figure 7 (section 3.2.) are not consistent. Also it is not clear how lodging area is estimate by nDSM. Adding a legend for the color-coded areas would be helpful.
Author Response
Dear Reviewer:
Thank you for your letter and for the reviewers’ comments concerning our manuscript entitled “Phenotyping of corn plants using unmanned aerial vehicle (UAV) images” (Ref: remotesensing-553815). Your comments are all valuable and very helpful for revising and improving our paper, as well as the important guiding significance to our researches. We have added some description about the methods, including GLCM, PROSAIL model, corn plant heights estimation etc. Moreover, the introduction has been improved. In the rest of the letter, we have provided specific responses to each review comment.
We are grateful for the review comments, and we hope that our responses and the resulting changes in the manuscript will be satisfactory. However, we will be happy to work with the editor and reviewers to resolve any remaining issues. Sincerely,
Wei Su, Mingzheng Zhang, Dahong Bian, Zhe Liu, Jianxi Huang, Wei Wang, Jiayu Wu, Hao Guo
Reply to comments:
1. Although the paper is well organized, it omits some important description about the methods used in the research e.g. GLCM,...and refers reader to find about the methods in references. It would be helpful if some basic information about the methods be included.
Reply:① the descriptions about GLCM computation, features and application essentials were added from Line 259 to Line 275. ② Moreover, the forward simulation steps of PROSIAL model has been added from Line 317 to Line 325. ③ the UAV image pre-processing were added from Line 186 to Line 187, Line 194 to Line 195, Line 198 to Line 199. ④ the selecting bare ground points, interpolation and calculation of difference between DSM and DTM were completed using our custom code, which were added from Line 247 to Line 249. ⑤ the height difference was determined by the threshold which were captured by statistics of lodging and non-lodging areas, which has been added from Line 277 to Line 278.
2. The contribution of this paper is not clear. There is a huge literature to estimate and study corn height using SFM and texture analysis. What is the contribution of your research compared with your pervious work and literature?
Reply: yes, there are huge researches about corn height estimation using SfM method. Our contributions lie in two aspects: (1) the height estimation results can be used to find the corn varieties with optimal plant heights and lodging resistance. (2) the CV of plant heights were analyzed to depict the corn canopy uniform and fluctuation, which was used to validate the stability of a corn variety. The contribution has been added in discussion from Line 499 to Line 503.
3. How many GCPs were used and how was their distribution?
Reply: There were 21 GPS control points in Boagaofeng Farm, and 36 GPS control points in Shunyi corn breeding trials. There were two principles for GPS distribution: (1) These GPS points were distributed throughout the study area as evenly as possible. (2) the GPS points were commonly selected on the obvious objects. These descriptions have been added from Line 175 to Line 179.
4. For radiometric calibration, at what height/altitude the pictures were taken from the test pattern?
Reply: for the radiometric calibration, the Sequoia drone was placed over the test pattern hold by operator with the height about 1m above ground, which was depicted from Line 194 to Line 195.
5. Line 218: how did you calculate the local maximum height of the nDSM? Using a n*n window? Local maximum can be a noise.
Reply: yes, indeed. So the corn plant height was estimated using the height of the nDSM pixel by pixel in this study firstly, not computing the individual plants one by one. Secondly, the corn plant heights in every corn breeding trial were analyzed statistically. One reason lies in this is reducing the effects of noise. In addition, this is corn breeding experiment, aiming at finding the overall gene performance, not only individual plant.
6. Line 246: “.. for the SAIL model to retrieve LAI using canopy inputs such as LAI, the leaf angle….” . should it be “ such as measured LAI, ..” ?
Reply: yes, thank you very much for pointing out this confusion. And the followed LAI, the leaf angle ext. are measured data used as prior knowledge. This has been added now.
7. Captions and description of figure 7 (section 3.2.) are not consistent. Also it is not clear how lodging area is estimate by nDSM. Adding a legend for the color-coded areas would be helpful.
Reply: yes, thank you very much for pointing out this inconsistent. (1) the caption and description of figure 7 are unified now, and (a) is the pre-processed RGB UAV image, (b) is the spatial distribution based on the ASM texture feature, (c) is the estimated lodging area based on the ASM texture feature, (d) is the estimated lodging area using the nDSM, (e) is the proportion of the lodging area in every plot, and (f) is the coefficient of variation (CV) of the lodging area in every plot. In addition, the description of Figure 7 (a) was added from Line 388 to Line 390. (2) the description about how lodging area is estimate by nDSM was added from Line 278 to Line 279. Furthermore, the description about how lodging area is estimate by ASM was added from Line 271 to Line 276. (3) the labels of field No. 4, field No. 6, field No. 8, lodging area and non-lodging area were added, too. And the legends of lodging area and non-lodging area in Figure 7 (c) and Figure 7 (d) were used to label the lodging area and non-lodging area.

Round 2
Reviewer 1 Report
The manuscript is much improved from the previous version. I just have one concern.
1. The authors estimated the height for each pixel in this study, and they did not estimate the plant height for individual plants. I am still confused about the plant height estimation. How did the authors estimate plant height for the 28 corn plants to access the height estimation accuracy? According to Figure 5, I think there was only one estimated value for each plant.
Author Response
Dear Reviewer:
Thank you for your affirmation about last reply about our manuscript entitled “Phenotyping of corn plants using unmanned aerial vehicle (UAV) images” (Ref: remotesensing-553815). Yes, it is. For the measure height of 28 corn plants used for estimation accuracy assessment, there was only one estimated value for each plant and the heights were measured using a meter stick from the ground to the tip of the corn plants. The measured corn plants were located using a Huace i80 real-time kinematic (RTK) GPS receiver. During the height estimation accuracy assessment, this height was compared with the height of pixel on this location. This has been updated and improved from Line 229 to Line 232, Line 368 to Line 370. Moreover, we improved the methods description, which could be found from Line 256, Line 261, Line 266, Line 276 to Line 280, Line 331 to Line 337, Line 342 to Line 344.
We are grateful for the review comments, and we hope that our responses and the resulting changes in the manuscript will be satisfactory. Sincerely,
Wei Su, Mingzheng Zhang, Dahong Bian, Zhe Liu, Jianxi Huang, Wei Wang, Jiayu Wu, Hao Guo
